# Conductive 2D metal-organic framework for high-performance cathodes in aqueous rechargeable zinc batteries

Kwan Woo Nam [1,7], Sarah S. Park [1,7], Roberto dos Reis [2,3], Vinayak P. Dravid [2,3], Heejin Kim [4], Chad A. Mirkin [1] & J. Fraser Stoddart [1,5,6]*

Currently, there is considerable interest in developing advanced rechargeable batteries that boast efficient distribution of electricity and economic feasibility for use in large-scale energy storage systems. Rechargeable aqueous zinc batteries are promising alternatives to lithium-ion batteries in terms of rate performance, cost, and safety. In this investigation, we employ $Cu_3(HHTP)_2$, a two-dimensional (2D) conductive metal-organic framework (MOF) with large one-dimensional channels, as a zinc battery cathode. Owing to its unique structure, hydrated $Zn^{2+}$ ions which are inserted directly into the host structure, $Cu_3(HHTP)_2$, allow high diffusion rate and low interfacial resistance which enable the $Cu_3(HHTP)_2$ cathode to follow the intercalation pseudocapacitance mechanism. $Cu_3(HHTP)_2$ exhibits a high reversible capacity of 228 mAh g$^{-1}$ at 50 mA g$^{-1}$. At a high current density of 4000 mA g$^{-1}$ (~18 C), 75.0% of the initial capacity is maintained after 500 cycles. These results provide key insights into high-performance, 2D conductive MOF designs for battery electrodes.

[1] Department of Chemistry, Northwestern University, Evanston, IL 60208, USA. [2] Department of Materials Science and Engineering, Northwestern University, Evanston, IL 60208, USA. [3] Northwestern University Atomic and Nanoscale Characterization Experimental (NUANCE) Center, Northwestern University, Evanston, IL 60208, USA. [4] Electron Microscopy Research Center, Korea Basic Science Institute, 169-148 Gwahak-roYuseong-guDaejeon 34133, Republic of Korea. [5] Institute for Molecular Design and Synthesis, Tianjin University, 92 Weijin RoadNankai DistrictTianjin 300072, China. [6] School of Chemistry, University of New South Wales, Sydney, NSW 2052, Australia. [7] These authors contributed equally: Kwan Woo Nam, Sarah S. Park.
*email: stoddart@northwestern.edu

Societal interest in energy storage systems (ESSs) has been increasing rapidly with the need to utilize and distribute effectively electricity generated using renewable energy sources[1–3]. Among the most suitable candidates for energy storage are lithium-ion batteries (LIBs) since they provide high performance in mobile devices, such as cellular phones and laptops. Their utilization, however, in large-scale applications, such as electric vehicles, is inhibited by high material costs and safety concerns[4,5]. In order to resolve the limitations of LIBs, numerous investigations[4–7] have been focused on greener electrode materials and aqueous electrolytes. From these perspectives, rechargeable aqueous zinc batteries (ZBs) have recently attracted[8–13] considerable attention for use in large-scale ESSs because of their high theoretical capacity (820 mAh g$^{-1}$), their low toxicity, and the relatively low cost of zinc[14]. Furthermore, ZBs operate in aqueous electrolytes[4,5], thereby gaining additional advantages related to safety, cost, and rate performance.

Despite all these advantages, rechargeable ZBs have several obstacles that need to be addressed before they can hope to replace LIBs in terms of electrochemical performance[15,16]. In particular, the development of a new high-performance cathode is crucial for the commercialization of ZBs. $\alpha$-MnO$_2$ with a $2 \times 2$ tunnel structure has been used[14] as a rechargeable ZB cathode, in which the large tunnels facilitate Zn$^{2+}$ ion diffusion within the host structure, providing high capacity and rate performance. These materials, however, are associated with low cyclability that can be attributed[15,16] to an unstable phase transition from a tunneled to a layered structure with simultaneous Mn$^{2+}$ dissolution during the discharge–charge process. Vanadium-based cathodes[8,17] also provide high capacity and rate performance, although the costliness of vanadium prohibits large-scale energy storage applications. Recently, organic-based cathodes, such as quinone derivatives, have been investigated because they are low cost, ubiquitous, and lightweight compared with inorganic cathodes[13,18]. Dissolution issues, however, during battery cycling inhibit the use of quinone derivatives in ZBs. In an effort to improve the stability of the quinone-based materials, polymerization[19], carbon composites[20], and an extended analog[13] have all been explored: the dissolution issues, however, of organic cathodes remain a drawback. Reflecting on all these difficulties, the development of new materials for ZB cathodes is a necessity.

Conductive metal-organic frameworks (MOFs) provide excellent platforms for resolving dissolution issues, related to organic-based cathodes. In these MOFs, the active organic species are immobilized by metal-ligand coordinate covalent bonds. In addition, their porous structures and electrical conductivities are favorable to ion and electron transport in the framework, improving high rate capability and cyclability. The potential applications of these materials in batteries has been confirmed, with high performance being achieved in electrochemical double-layer capacitors[21,22] and Na$^+$ storage[23], as well as in reports of their use in various battery systems[24–27].

We introduce the idea of utilizing a two-dimensional (2D) conductive MOF, Cu$_3$(HHTP)$_2$ (HHTP = 2,3,6,7,10,11-hexahydroxytriphenylene)[28], as the cathode material for rechargeable aqueous ZBs. Electrical conductivity (0.2 S cm$^{-1}$, four-point probe, single crystal)[28] and large pores (~2 nm) facilitate electron and Zn$^{2+}$ ion transport to active sites. In particular, we anticipate that the redox activity of the quinoid units of HHTP[28–30] with Zn$^{2+}$ insertion will promote the performance of the cathode.

Here, on account of these properties, we have tested the electrochemical performance of the Cu$_3$(HHTP)$_2$ cathode. Cu$_3$(HHTP)$_2$ shows redox switching at 1.06 V and 0.88 V vs. Zn/Zn$^{2+}$ with the highest reversible capacity of 228 mAh g$^{-1}$ at 50 mA g$^{-1}$ to the best of our knowledge. These reversible capacities in rechargeable aqueous ZBs are the first example in MOFs and one of the highest reported values for cathodes with open-framework structures, including Prussian Blue analogs[31–33] that have exhibited substantially smaller values of <70 mAh g$^{-1}$ at similar current densities. In addition, the high diffusion rate of Zn$^{2+}$ ions and low interfacial resistance by the insertion of hydrated Zn$^{2+}$ ions allows Cu$_3$(HHTP)$_2$ to follow the intercalation pseudo-capacitance mechanism. As a consequence, Cu$_3$(HHTP)$_2$ achieves a high rate performance and cyclability, indicating that 75.0% of the initial capacity (124.4 mAh g$^{-1}$) is maintained after 500 cycles at an extremely high current density of 4000 mA g$^{-1}$ (~18 C). This work reveals the reason for the observed high rate performance and charge-storage mechanism of the Cu$_3$(HHTP)$_2$, which is poised to facilitate the development of 2D conductive MOFs for energy storage.

## Results

**Synthesis and characterization of Cu$_3$(HHTP)$_2$.** Cu$_3$(HHTP)$_2$ was synthesized according to a previously reported procedure[28] and applied as the cathode material for aqueous rechargeable ZBs (Fig. 1a). PXRD analysis confirmed that the as-synthesized Cu$_3$(HHTP)$_2$ comprises (Fig. 1b) hexagonal 2D sheets stacked in a slipped-parallel configuration along the $c$ axis[29,34]. Cu$_3$(HHTP)$_2$ was indexed based on a hexagonal unit cell (Fig. 2a) with the space group $P6/mmm$. The lattice parameters were calculated to be $a = b = 21.2$ Å and $c = 6.6$ Å with Rietveld refinement ($R_p = 3.41$, $R_{wp} = 4.52$, $\chi^2 = 3.06$). The morphology of Cu$_3$(HHTP)$_2$ was also investigated by field-emission scanning electron microscopy (FE-SEM). The shape of Cu$_3$(HHTP)$_2$ is similar[28] (Fig. 2b and Supplementary Fig. 1a, b) to that of the uniform rods of Ni$_3$(HHTP)$_2$. The electrical conductivity of Cu$_3$(HHTP)$_2$ powder and Cu$_3$(HHTP)$_2$ electrode composite (60 wt% Cu$_3$(HHTP)$_2$, 20 wt% acetylene black, and 20 wt% PVDF) were measured on a pressed pellet using the two-point probe method. The conductivities obtained were 0.01 and 0.04 S cm$^{-1}$ for Cu$_3$(HHTP)$_2$ powder and electrode composite, respectively. The electrical conductivity of a bulk Cu$_3$(HHTP)$_2$ electrode matches well the previously reported values[29].

A transmission electron microscopy (TEM) image also reveals (Supplementary Fig. 2a) the one-dimensional (1D) nanorod structure of Cu$_3$(HHTP)$_2$. The length of the Cu$_3$(HHTP)$_2$ nanorods extends (Fig. 2b and Supplementary Fig. 1a, b, and 2a) a few micrometers with diameters of around 20–500 nm. In addition, a low dose—high resolution transmission electron microscopy (LD—HRTEM) image (Fig. 2d) enlarged from the selected yellow area in Fig. 2c (selected from Supplementary Fig. 2b) shows large pores with diameters of ~2.0 nm with a honeycomb arrangement viewed along the [001] direction. An enlarged LD-HRTEM image (Fig. 2g) from the selected area in Fig. 2e shows parallel Cu$_3$(HHTP)$_2$ nanorods along the [010] zone axis with a lattice distance of 2.0 nm for the (100) crystal plane. Fast Fourier transform (FFT) (Fig. 2f) from the selected area (Fig. 2e) indicates clearly that the Cu$_3$(HHTP)$_2$ nanorods have well developed (100) and (200) planes. These planes indicate[28] that the as-synthesized Cu$_3$(HHTP)$_2$ is highly crystalline in nature with the [100] axis being the preferred orientation for the 1D nanorods. The unique structure of Cu$_3$(HHTP)$_2$, along with the shape of the 1D nanorods and the large pores, facilitate the diffusion of Zn$^{2+}$ ions during the discharge–charge process. In addition, scanning electron microscopy-energy-dispersive X-ray spectroscopy (SEM-EDX) was used to verify (Supplementary Fig. 1c, d) the C, O, and Cu content of the Cu$_3$(HHTP)$_2$ particles.

**Electrochemical performance of Cu$_3$(HHTP)$_2$.** A cyclic voltammogram of Cu$_3$(HHTP)$_2$ thin film on SUS foil in a 3.0 M

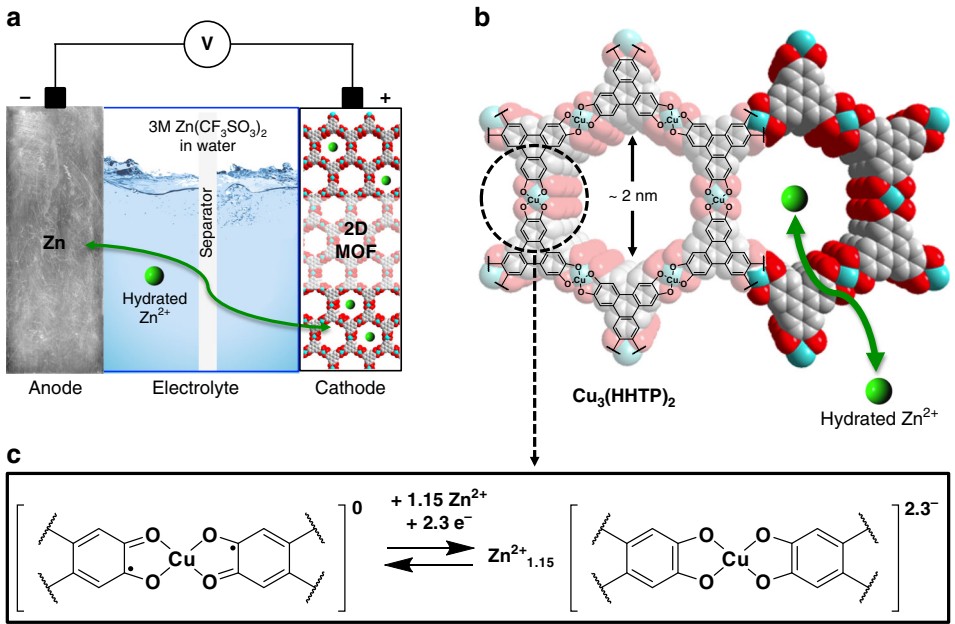

**Fig. 1** Zn-Cu$_3$(HHTP)$_2$ chemistry. **a** Schematic illustration of the rechargeable Zn-2D MOF cell. **b** Structure of Cu$_3$(HHTP)$_2$, which when viewed down the $c$ axis, exhibits slipped-parallel stacking of 2D sheets with a honeycomb lattice. The cyan, red, and gray spheres represent Cu, O, and C atoms, respectively. The H atoms are omitted for the sake of clarity. **c** Expected redox process in the coordination unit of Cu$_3$(HHTP)$_2$

aqueous solution of Zn(CF$_3$SO$_3$)$_2$ indicates (Supplementary Fig. 3) that the Zn$^{2+}$ insertion and extraction reaction is reversible. The reaction of Zn$^{2+}$ ions with Cu$_3$(HHTP)$_2$ occurs reversibly at approximately 0.65/1.10 V and 0.90/1.21 V (vs. Zn/Zn$^{2+}$), respectively. Galvanostatic tests revealed that this reversibility is reflected (Fig. 3a) in the voltage profiles, with plateaus at the corresponding voltages. The first discharge plateau at ~0.90 V (vs. Zn/Zn$^{2+}$) originates from the redox process between Cu$^{2+}$ and Cu$^{+}$. Furthermore, the second discharge plateau at 0.65 V (vs. Zn/Zn$^{2+}$) may be attributed to a two-electron uptake associated with the HHTP linkers. A detailed redox reaction mechanism is discussed in the upcoming sections. The initial reversible capacity is 228 mAh g$^{-1}$ at a rate of 50 mA g$^{-1}$, followed by a capacity of 215 mAh g$^{-1}$ in the second cycle, and the voltage profiles and capacity are retained (Fig. 3a and Supplementary Fig. 4) for 30 cycles. These reversible capacities are quite remarkable, providing some of the highest reported values for cathodes with open-framework structures, including Prussian Blue analogs[31–33], that have been applied (Supplementary Table 1) to aqueous rechargeable ZBs.

In order to verify the influence of the Cu$_3$(HHTP)$_2$ 2D structure with large pores on the electrochemical performance, we conducted rate-capability tests. In these electrochemical tests, Cu$_3$(HHTP)$_2$ demonstrated excellent rate capability (Fig. 3b). The Cu$_3$(HHTP)$_2$ electrode exhibited capacities of 191.4, 189.2, 152.4, and 124.5 mAh g$^{-1}$ when the current density was increased, respectively, by 2, 4, 10, and 80 times (100, 200, 500, and 4000 mA g$^{-1}$) from 50 mA g$^{-1}$. These results correspond to capacity retentions of 89.0%, 88.0%, 70.9%, and 57.9%, respectively, with respect to the initial capacity of 215.0 mAh g$^{-1}$. Moreover, the Cu$_3$(HHTP)$_2$ electrodes show promising cycling stability. At a current density of 500 mA g$^{-1}$ (~2 C), 75.0% of the initial capacity (152.5 mAh g$^{-1}$) was maintained (Fig. 3c) after 100 cycles. In addition, by increasing the mass loading of active materials from 60 to 90%, although the initial capacity decreased slightly to 125 mAh g$^{-1}$ at 500 mA g$^{-1}$ (Supplementary Fig. 5a), the retention of capacity after 100 cycles was 76% of the initial capacity (Supplementary Fig. 5b). This capacity retention for

a 90% active materials loading electrode is almost identical to that of a 60% active materials loading electrode. Furthermore, at an extremely high current density of 4000 mA g$^{-1}$ (~18 C), 75.0% of the initial capacity (124.4 mAh g$^{-1}$) was maintained (Fig. 3d) after 500 cycles. This cyclability reflects the structural stability of Cu$_3$(HHTP)$_2$ during repeated (de)intercalation of the Zn$^{2+}$ ions.

**Origin of high rate performance of Cu$_3$(HHTP)$_2$.** In order to investigate more detailed reasons for the high rate performance of Cu$_3$(HHTP)$_2$, diffusion coefficient and interfacial resistance studies were carried out. The Zn$^{2+}$ ion diffusion coefficient of Cu$_3$(HHTP)$_2$ was obtained by applying galvanostatic intermittent titration technique (GITT) measurements; See the Supplementary Note 1 for details. The overall diffusion coefficient of Zn$^{2+}$ ions in Cu$_3$(HHTP)$_2$ over the whole potential range was $3.9 \times 10^{-10}$ cm$^2$ s$^{-1}$ (Supplementary Fig. 6), which is similar to that of single crystalline Zn$_{0.25}$V$_2$O$_5 \cdot n$H$_2$O nanobelts[8]. Specifically, by excluding the loss of diffusion coefficient from the high overpotential of the copper redox region, attributed to the self-discharge, and calculating the diffusion coefficient only with the main redox region of the quinoid, the diffusion coefficient of Zn$^{2+}$ ions in Cu$_3$(HHTP)$_2$ showed $1.2 \times 10^{-9}$ cm$^2$ s$^{-1}$ (Supplementary Fig. 6), indicating fast redox reactions.

Furthermore, an interfacial resistance between the electrode and the electrolyte was studied in order to determine the rate performance of electrode materials. In order to investigate the interfacial resistance of the Cu$_3$(HHTP)$_2$ electrode, an electrochemical impedance spectroscopy (EIS) investigation was conducted with symmetric cells of the Cu$_3$(HHTP)$_2$ electrodes in aqueous or organic electrolytes. Notably, the interfacial resistance of Cu$_3$(HHTP)$_2$ showed 150 and 16,000 Ω cm$^2$ (Supplementary Fig. 7), and obtained conductivities from these interfacial resistances of Zn$^{2+}$ ions are $0.7 \times 10^{-2}$ and $0.6 \times 10^{-5}$ S cm$^{-1}$ (Supplementary Fig. 7) in aqueous and organic electrolytes, respectively. In recent studies, the insertion of carrier ions with H$_2$O molecules has been suggested as the reason for low interfacial resistance, because the H$_2$O can decrease the

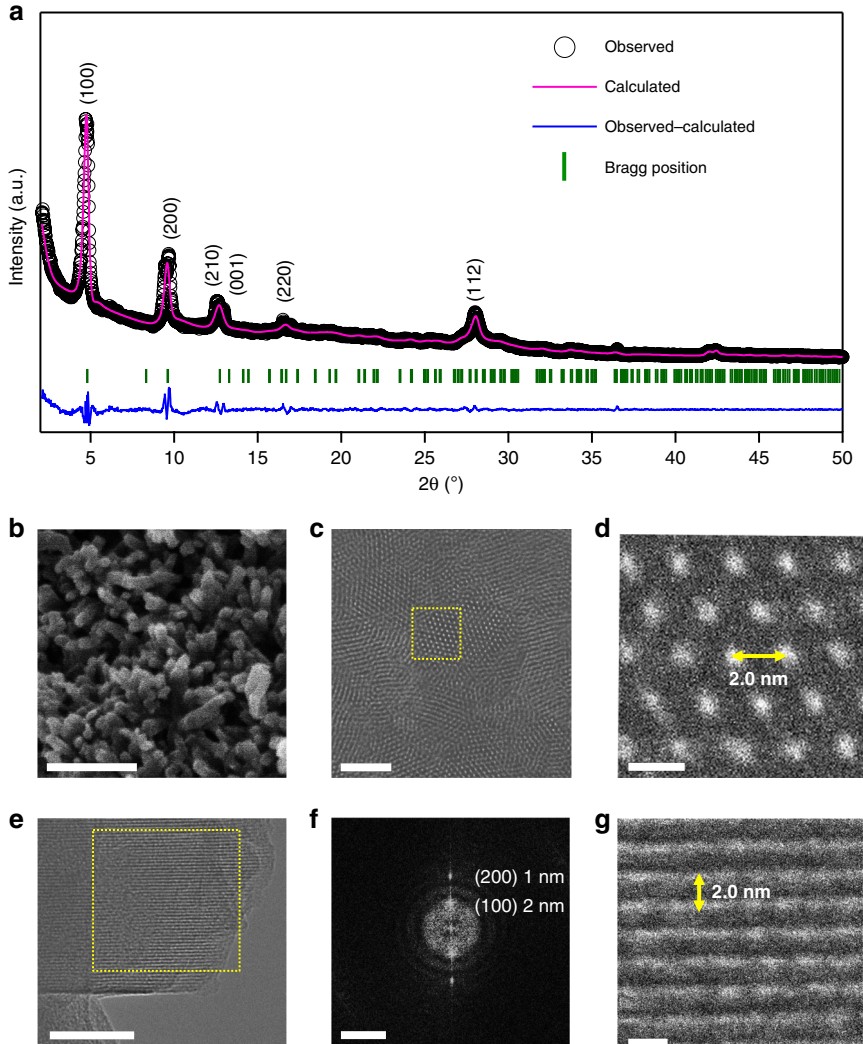

**Fig. 2** 2D Chemical structure and structural analysis of $Cu_3(HHTP)_2$. **a** Rietveld refinement of PXRD patterns. **b** FE-SEM image of $Cu_3(HHTP)_2$, scale bar: 200 nm. **c** LD-HRTEM image of $Cu_3(HHTP)_2$ at a low resolution, scale bar: 20 nm. **d** LD-HRTEM image of $Cu_3(HHTP)_2$ along the [001] zone axis, indicating a hexagonal pore packing with $d_{100} = 2.0$ nm, scale bar: 2 nm. **e-g** LD-HRTEM images at (**e**) low and (**g**) high resolution along the [010] direction. Scale bars in (**e**) and (**g**) are 50 and 2 nm, respectively. **f** An FFT pattern of the yellow square in (**e**), scale bar: 2 nm$^{-1}$

desolvation energy[35] and the Coulombic repulsion from the interface[36–38]. Existence of $H_2O$ in the discharged electrode was tested with thermogravimetric analysis (TGA). The TGA profile of the discharged electrode showed a 26.9% weight loss between 120 and 300 °C (Supplementary Fig. 8), indicating that the low interfacial resistance of $Cu_3(HHTP)_2$ in the aqueous electrolyte can be attributed to the insertion of $H_2O$ with $Zn^{2+}$ ions during the discharge reaction. We assume the large pore size of $Cu_3(HHTP)_2$ supports the insertion of hydrated $Zn^{2+}$ ions. In order to confirm the importance of $H_2O$, the performance of $Cu_3(HHTP)_2$ was studied in an organic electrolyte. On account of the high interfacial resistance caused by the organic electrolyte, the initial discharge capacity of $Cu_3(HHTP)_2$ decreased to 144 mAh g$^{-1}$ at a rate of 50 mA g$^{-1}$, and the subsequent charging reaction almost did not occur because of the high overpotential (Supplementary Fig. 9a). This phenomenon which is also evident[35–37] in the case of $Mg^{2+}$ ions in organic electrolytes, is caused by the strong interaction of divalent ions with the cathode, i.e., extracting of divalent ions from the host electrode is unfavorable[35–37]. As a result of this phenomenon, the capacity retention of $Cu_3(HHTP)_2$ in the organic electrolyte was almost zero (Supplementary Fig. 9b). In total, the origin of the high rate

properties of $Cu_3(HHTP)_2$ is thought to be a consequence of the high diffusion rate of $Zn^{2+}$ ions in the cathode and low interfacial resistance by the hydrated $Zn^{2+}$ ion insertion.

**Electronic states analysis during discharge–charge**. With a view to investigating changes in the electronic states of $Cu_3(HHTP)_2$ during discharge-charge, X-ray photoelectron spectroscopy (XPS) was conducted on the Zn, O, and Cu elements. After inserting $Zn^{2+}$ ions into the $Cu_3(HHTP)_2$, the Zn 2$p$ peaks appear and disappear (Fig. 4a and Supplementary Fig. 10) at the discharged and charged states, respectively; this behavior is a consequence of the reversible insertion/extraction of $Zn^{2+}$ into/from the $Cu_3(HHTP)_2$ cathodes. The quinoid peak at 532 eV shifts (Fig. 4b) to a benzoid peak at 533 eV in the O 1$s$ spectrum, while discharging from 0.8 V (point b in Fig. 3a) to a fully discharged state (point c in Fig. 3a). The peaks which had shifted returned to their original positions, while charging from the fully discharged state (point c in Fig. 3a) to 1.15 V (point d in Fig. 3a). This shift reveals that the second plateau (Fig. 3a), which exists during the discharge process, originates from the quinoid structure acting as a redox center. Based on these XPS results, we infer that the

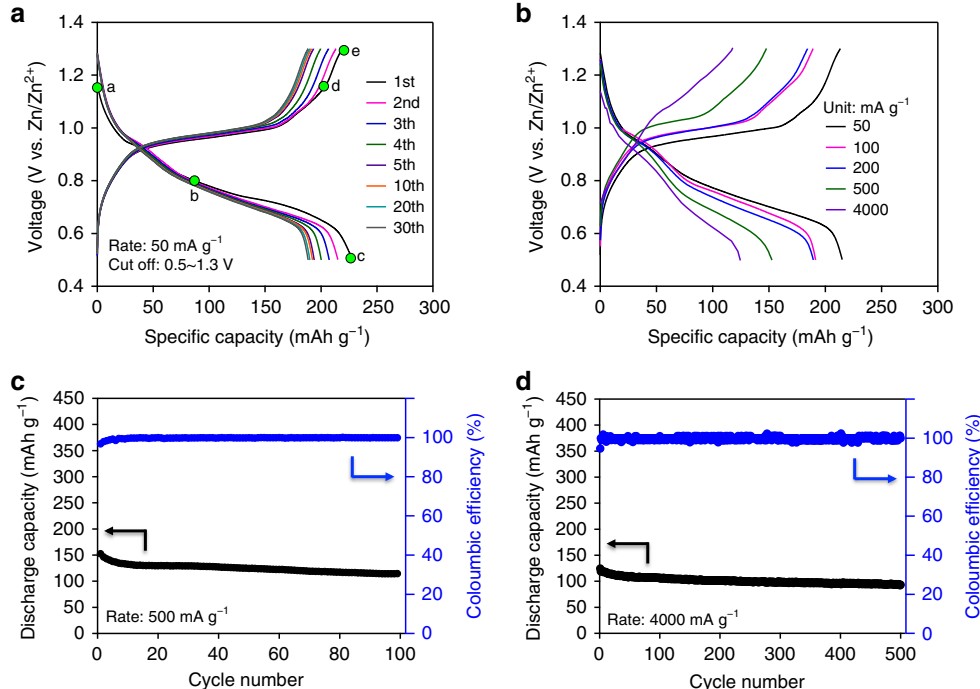

**Fig. 3 Electrochemical performance of Cu$_3$(HHTP)$_2$. a, b** Discharge–charge voltage profiles of Cu$_3$(HHTP)$_2$ at **a** 50 mA g$^{-1}$ and **b** various current densities. The green dots labeled with (a-e) in (**a**) are states where XPS analysis in Fig. 4b, c was conducted. **c, d** Cycling performance of Cu$_3$(HHTP)$_2$ at current densities of **c** 500 mA g$^{-1}$ and **d** 4000 mA g$^{-1}$

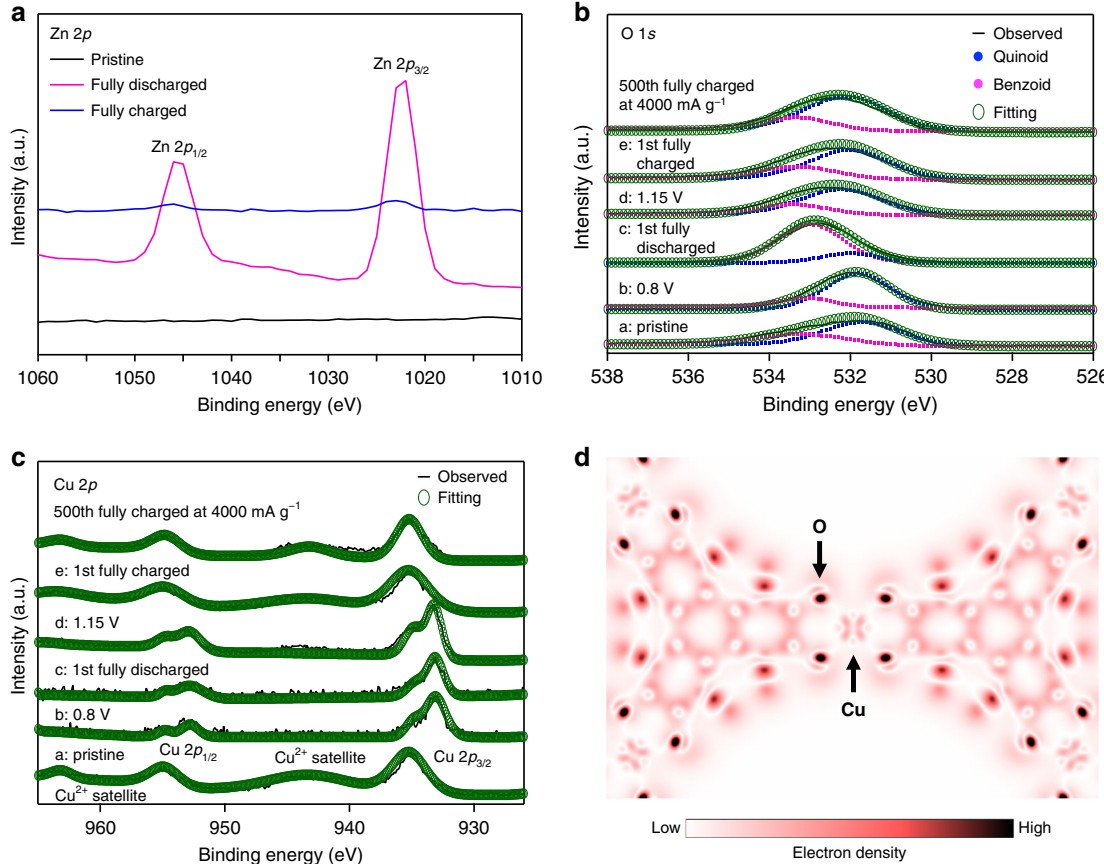

**Fig. 4 Electronic states analysis during discharge–charge. a–c** Ex situ XPS spectra of **a** Zn 2$p$, **b** O 1$s$, and **c** Cu 2$p$. **d** Changes of electron density upon the reduction of Cu$_3$(HHTP)$_2$

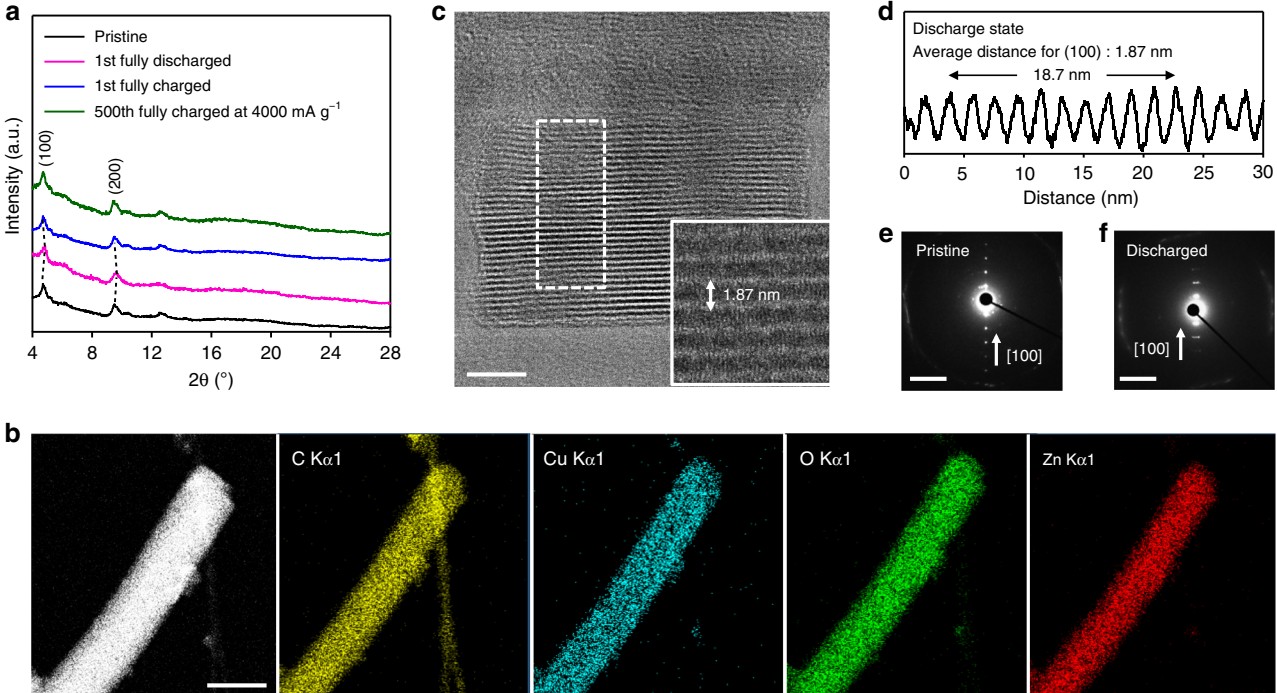

**Fig. 5** Structure analysis during discharge–charge. **a** PXRD patterns of the $Cu_3(HHTP)_2$ electrode in the pristine, first fully discharged/charged states at a rate of 50 mA g$^{-1}$, and 500th fully charged states at a rate of 4000 mA g$^{-1}$. **b** Scanning transmission electron microscopy (STEM) image of the fully discharged $Cu_3(HHTP)_2$ alongside its EDX elemental mapping with respect to C, Cu, O, and Zn, suggesting uniform Zn insertion over the electrode, scale bar: 100 nm. **c** An LD-HRTEM image of discharged $Cu_3(HHTP)_2$ viewed down the [010] zone axis. An inset in (**c**) shows a magnified area depicting the (100) plane, scale bar: 20 nm. **d** Measurements of the (100) interplanar distances from the white boxed area in (**c**) indicate the average $d_{100} = 1.87$ nm. **e**, **f** SAD patterns from $Cu_3(HHTP)_2$ at (**e**) pristine and (**f**) discharged states used to confirm the interplanar distances of (100). The arrows and scale bar indicate the [100] direction and 2 nm$^{-1}$, respectively

quinoid structure is involved in the redox reaction; a similar redox mechanism was reported[26] for Cu(2,7-AQDC) MOF (2,7-$H_2$AQDC = 2,7-anthraquinonedicarboxylic acid), where oxygen and copper are the redox centers for LIBs. Similarly, the presence of transition metals involved in the redox reaction in our system causes the peaks of $Cu^{2+}$ satellites in the pristine state to disappear (Fig. 4c). The Cu 2$p$ peaks then separate into lower binding-energy peaks between the pristine state (point a in Fig. 3a) and 0.8 V (point b in Fig. 3a) in the Cu 2$p$ spectrum (Fig. 4c). There is then no further shift in the Cu 2$p$ peaks that lie between 0.8 V (point b in Fig. 3a) and 1.15 V (point d in Fig. 3a). As expected, the initial Cu 2$p$ spectrum was reinstated, including its original profiles, between 1.15 V (point d in Fig. 3a) and the fully charged state (point e in Fig. 3a). From these changes in the Cu 2$p$ peaks, the first plateau (Fig. 3a) that appears during the discharge process can be attributed to a partial redox reaction from $Cu^{2+}$ to $Cu^+$. Consequently, these XPS analyses suggest that both the quinoid component and the copper in $Cu_3(HHTP)_2$ participate as redox centers during the discharge–charge process. The theoretical capacity of $Cu_3(HHTP)_2$ should be 197 mAh g$^{-1}$, when using the quinoid structure as the redox center and inserting $Zn^{2+}$ ions with two electrons. The initial capacity determined (Fig. 3a), however, for $Cu_3(HHTP)_2$ is 228 mAh g$^{-1}$, revealing that these $Cu_3(HHTP)_2$ cathodes can obtain 2.3 electrons (Fig. 1c). In light of these XPS results, the additional discharge capacity of $Cu_3(HHTP)_2$, equivalent to 0.3 electrons, can be derived from the redox events of $Cu^{2+}$. In order to identify the redox center of $Cu_3(HHTP)_2$, density functional theory (DFT) calculations were performed. When supplying 6.9 extra electrons to $Cu_3(HHTP)_2$, Cu atom, as well as to the linker, takes of the additional electron (Fig. 4d), indicating that Cu atoms participate

in the reduction reaction; See the Supplementary Note 2 for details. In the density of states (DOS) analysis (Supplementary Fig. 11), the electronic states just above the Fermi level consist of O, C, and Cu. This result supports the observed redox events occurring at these atoms. Furthermore, both peaks of O 1$s$ and Cu 2$p$ of the charged electrode after 500 cycles at a rate of 4000 mA g$^{-1}$ (Fig. 4b, c) are more or less similar to those of the pristine electrode, indicating that the redox reaction of $Cu_3(HHTP)_2$ is highly reversible.

**Structure analysis during discharge–charge**. The PXRD patterns of $Cu_3(HHTP)_2$ in the discharged (inserting $Zn^{2+}$ ions into $Cu_3(HHTP)_2$) electrode demonstrate that the (100) peak has a slight right-side shift from 4.70° to 4.85°, revealing (Fig. 5a) that the pore size in $Cu_3(HHTP)_2$ decreases from 19.3 to 18.7 Å. This change indicates that inserting $Zn^{2+}$ ions into $Cu_3(HHTP)_2$ decreases the pore size of $Cu_3(HHTP)_2$ as a result of the electrostatic interaction between divalent $Zn^{2+}$ cations and the oxygen anion of the host structure. With the exception of peak shifts following $Zn^{2+}$ insertion, no changes (appearance or disappearance of peaks) are observed, indicating that the discharge process does not include $H^+$ insertion accompanied by the formation of the $Zn(OH)_2$ analog[39]. After the charge process (extracting $Zn^{2+}$ ions from $Cu_3(HHTP)_2$), the PXRD peaks in the charged electrode return fully (Fig. 5a) to the position of the original pristine state. In addition, after 500 cycles at a rate of 4000 mA g$^{-1}$, the PXRD patterns of $Cu_3(HHTP)_2$ are identical (Fig. 5a) to those of the pristine state. This observation implies that the inserted $Zn^{2+}$ ions only affect the pore size of the host structure and that the structure of $Cu_3(HHTP)_2$ is maintained

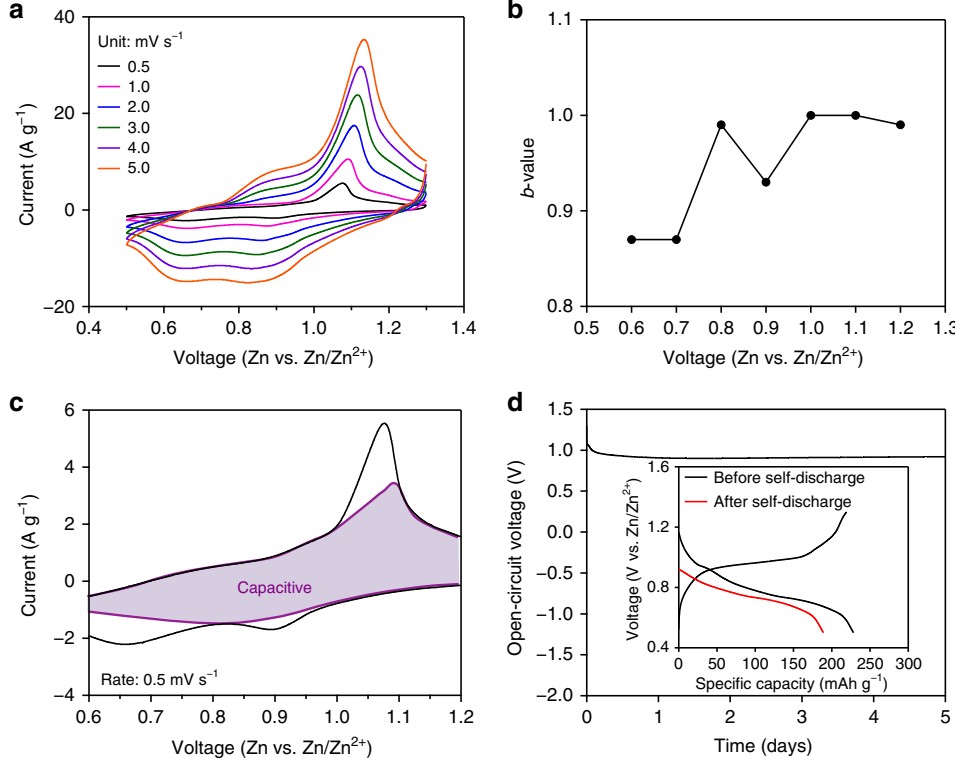

**Fig. 6** Charge-storage mechanism of $Cu_3(HHTP)_2$. **a** Cyclic voltammograms of $Cu_3(HHTP)_2$ recorded at different scan rates. **b** $b$-values for the $Cu_3(HHTP)_2$ electrodes plotted as a function of the potential for cathodic scans. **c** Capacitive and diffusion currents contributed to the charge-storage of $Cu_3(HHTP)_2$ at the rate of 0.5 mV s$^{-1}$. **d** A self-discharge profile of $Cu_3(HHTP)_2$. The inset shows voltage profiles for the self-discharge test before and after storage

robustly when $Zn^{2+}$ ions are inserted/extracted into/from $Cu_3(HHTP)_2$. Similarly, the morphology of the $Cu_3(HHTP)_2$, after $Zn^{2+}$ ion insertion (Supplementary Fig. 12b, d), is almost the same (Supplementary Fig. 12a, c) as that of $Cu_3(HHTP)_2$ in a pristine state. In addition, ion-exchange from $Cu^{2+}$ to $Zn^{2+}$ ions is endothermic by 0.8 eV per ion, according to DFT calculations (Supplementary Fig. 13), indicating the high stability of $Cu_3(HHTP)_2$ against $Zn^{2+}$ substitution. Consequently, the PXRD results lead us to infer that the $Zn^{2+}$ ions are accommodated in the large pores of $Cu_3(HHTP)_2$, thus enabling high long-term stability while cycling at a high rate.

**Confirmation of inserting $Zn^{2+}$ ions into the pore structure of $Cu_3(HHTP)_2$.** The uniform presence of $Zn^{2+}$ ions in $Cu_3(HHTP)_2$ nanorods was confirmed (Fig. 5b) by EDX chemical mapping which shows uniform distribution of Zn ions over the entire electrode area at the fully discharged state. In order to elucidate the consequences of the insertion of $Zn^{2+}$ ions into the pores of $Cu_3(HHTP)_2$, the lattice parameter changes were analyzed (Fig. 5c) with LD-HRTEM in the discharged state. Significantly, after inserting $Zn^{2+}$ ions into $Cu_3(HHTP)_2$ nanorods, the lattice distance of the (100) plane (inset of Fig. 5c, d) decreases slightly to 1.87 nm, demonstrating the same tendency observed (Fig. 5a) in the PXRD patterns. In addition, selected-area diffraction patterns from pristine and discharged samples (Fig. 5e, f) demonstrate that the (100) lattice distance decreases from 2.01(±0.01) nm to 1.90(±0.01) nm, in the consequent interaction of divalent cations inserted into the pores of the framework. This result verifies the fact that $Zn^{2+}$ ions are inserted into the pores in MOFs in a battery system.

**Charge-storage mechanism of $Cu_3(HHTP)_2$.** In order to understand the charge-storage mechanism of $Cu_3(HHTP)_2$, CV measurements were carried out using various scan rates (Fig. 6a). Currents depending on the scan rates study enables determining $b$-values from the equation of a power law[40–45]: $i = av^b$ where $i$ is the current (A), $v$ is the potential scan rate (V s$^{-1}$), $a$ and $b$ are arbitrary coefficients. Generally, battery electrode materials are characterized by $b = 0.5$, indicating a semi-infinite diffusion process[40–45], whereas the closer the $b$-values are to 1, the closer to the capacitive contribution. The $b$-values are the slope obtained by plotting the peak currents ($i$) and scan rates ($v$) in a log plot (Supplementary Fig. 14a) with an assumption that the current obeys the power-law relationship. The $b$-values of $Cu_3(HHTP)_2$ are above 0.85 within all operating voltage ranges (Fig. 6b), indicating the operating mechanism is not dominated by diffusion.

Furthermore, for quantitative analysis of capacitance, the scan rate dependence of the current was plotted (Supplementary Fig. 14b). The capacitive effect ($k_1v$) and diffusion-controlled insertion ($k_2v^{1/2}$) could be calculated with the plot, see the Supplementary Note 3 for details. The capacitive contribution was 83% (Fig. 6c) out of the total current, at a scan rate of 0.5 mV s$^{-1}$, indicating the total energy storage in $Cu_3(HHTP)_2$ arises from a capacitive process rather than the solid-state diffusion of $Zn^{2+}$ in $Cu_3(HHTP)_2$. Unlike a non-Faradaic surface adsorption present in the typical responses of a capacitor, reversible redox peaks on CV profiles (Fig. 6a, c) and the reversible shifts of (100) peaks in PXRD (Fig. 5a) during the discharge–charge process were observed, indicating that the $Cu_3(HHTP)_2$ follows an intercalation pseudocapacitance charge-storage mechanism[41–45]. In this mechanism, charge-storage occurs by intercalation/de-intercalation of cations in the bulk active materials, and its

kinetics are not limited by the diffusion of the cations. As a consequence, the advantage of batteries (high capacitance) and supercapacitors (high rate) are integrated into one system. In addition, a self-discharge test[46] was carried out to confirm the ability of charge-storage, and $Cu_3(HHTP)_2$ showed (Fig. 6d) remarkably low self-discharge rate of 0.003 V h$^{-1}$. The loss of the capacity during self-discharge mainly occurs near the Cu redox region above 0.9 V, agreeing well with the GITT study (Supplementary Fig. 6). Furthermore, after 5 days of storage, 83% of the initial capacity was still maintained (inset of Fig. 6d) and therefore proving its outstanding stability in the fully charged state.

## Discussion

In summary, we have demonstrated a $Cu_3(HHTP)_2$ 2D conductive MOF that may be utilized as a ZB cathode. The solid-state structure of $Cu_3(HHTP)_2$, with a high diffusion rate of $Zn^{2+}$ ions, and low interfacial resistance caused by the insertion of hydrated $Zn^{2+}$ ions, as a result of the large open channel structures, provides an increased rate performance and cyclability compared with those of conventional organic-based materials. In addition, the kinetic analyses of the electrochemical behavior of $Cu_3(HHTP)_2$ obtained by CV suggest that the charge-storage mechanism of $Cu_3(HTTP)_2$ is intercalation pseudocapacitance, indicating that the mechanism is not determined by diffusion. Furthermore, XPS measurements and DFT calculations suggest that $Cu_3(HHTP)_2$ utilizes both copper and the quinoid structure as redox-active sites, increasing the specific capacity of the material. In addition, the PXRD and LD-HRTEM data indicate that inserted $Zn^{2+}$ ions are stored in the $Cu_3(HHTP)_2$ pores. These findings point to the potential of these cathodes for use in large-scale applications. This investigation paves the way for the further exploration of 2D conductive MOFs with other transition metals that could increase their redox potential, thus improving the performance of 2D conductive MOF-based ZB cathodes.

## Methods

**Materials**. All commercially available reagents and solvents were purchased from Sigma-Aldrich and used as received without further purification. Zn and SUS films were purchased from Goodfellow. All the parts for making coin cells were obtained from Pred Materials International. $Cu_3(HHTP)_2$ was prepared according to a previously reported procedure[28], washed with deionized $H_2O$ and $Me_2CO$, respectively, and dried in air.

**Characterization**. The morphology of powder and elementary analysis was obtained by field-emission scanning electron microscopy (FE-SEM, Hitachi S-4800) with implemented energy-dispersive X-ray spectroscopy (EDX, Oxford Aztec X-max 80 SDD EDX detector). Images were acquired at a working distance of 7 mm with an electron beam energy of 20 kV and emission current of 20 μA. In order to investigate the $H_2O$ content after the discharge process, thermogravimetric analysis (TGA, Netzsch Jupiter) was performed by raising the temperature from room temperature to 300 °C at a ramp rate of 5 °C min$^{-1}$ under an Ar flow. Powder X-ray diffraction (PXRD, STOE STADI-P) with Cu-Kα1 radiation was measured through transmission geometry for crystal structure analysis by scanning in the 2θ range of 2°–90° with scan steps of 0.015° with accelerating voltage and current of 40 kV and 40 mA. For the characterization of $Cu_3(HHTP)_2$ at different charge and discharge states, the cells were opened and rinsed with deionized $H_2O$ inside a glove-box. The oxidation states of electrodes were analyzed by X-ray photoelectron spectroscopy (XPS, Thermo scientific ESCALAB 250Xi). Each sample was dried under vacuum for 1 h prior to XPS measurements. For the ex situ XPS characterization of $Cu_3(HHTP)_2$ at different charge and discharge states, the cells were opened and rinsed with deionized $H_2O$ inside a glove-box. The electrical conductivity of $Cu_3(HHTP)_2$ was measured by the two-point probe method at 25 °C. A pellet was placed on a home-built in situ pellet press[47] and connected to an electrometer (Keithley 4200-SCS). The current-voltage ($I–V$) measurements were performed at 25 °C by sweeping the voltage.

**Transmission electron microscopy**. Pristine and discharged $Cu_3(HHTP)_2$ MOF samples were dispersed in EtOH and drop-cast on lacey carbon Mo-based TEM grids. LD-HRTEM was performed using a JEOL Grand ARM instrument operated at 300 kV. Data were collected using a Gatan K3-IS direct electron detector. In

order to avoid MOF structure degradation under electron beams, images were collected at dose rates below 20 e$^-$/pixel/s and the cumulative dose in the range of 15–20 e$^-$/A[2 48]. For selected-area diffraction (SAD), the electron beam was spread out and with data acquired at low magnification to avoid sample damage. SAD Patterns were collected using a Gatan OneView camera. EDX data were collected using an SDD EDX detector.

**Electrochemical tests**. In order to investigate the electrochemical performance of $Cu_3(HHTP)_2$ as a cathode in zinc batteries, coin cells with a two-electrode configuration—which comprise a $Cu_3(HHTP)_2$ cathode and a Zn-film anode (100 μm in thickness)—were assembled. The $Cu_3(HHTP)_2$ electrode was first of all prepared by making a slurry containing $Cu_3(HHTP)_2$:acetylene black:poly(vinylidene difluoride) (PVDF) in the ratio of 60:20:20 or 90:5:5 in 1-methyl-2-pyrrolidinone (NMP), respectively. The slurry was then cast onto stainless steel (SUS 304) foil, followed by drying at 70 °C in a vacuum oven. The mass loading of the active material in each electrode was 2 mg cm$^{-2}$. The electrolyte solution was 3 M and 0.25 M zinc trifluoromethanesulfonate ($Zn(CF_3SO_3)_2$) in deionized $H_2O$ and acetonitrile (MeCN), respectively. All cells were aged for 1 h prior to initiating electrochemical processes to ensure good soaking of the electrolyte solution into the electrodes. The cells were cycled in the voltage range of 0.5–1.3 V (vs. Zn/$Zn^{2+}$). All measurements were made at 25 °C using a battery tester (BST8-300-CST, MTI, USA). All galvanostatic measurements were recorded in the constant current mode (no constant voltage steps). CV was carried out using coin cells with a two-electrode configuration, which comprise the $Cu_3(HHTP)_2$ cathode and the Zn-film anode (Reference 600 potentiostat, Gamry Instruments, USA). EIS measurements were performed on symmetric cells over the frequency range of 0.01 Hz–1 MHz with an input voltage amplitude of 10 mV (Reference 600 potentiostat, Gamry Instruments, USA).

**DFT calculations**. These calculations were performed using the Perdew–Burke–Ernzhof (PBE) exchange-correlation functional[49] and the projector-augmented wave (PAW) method[50] as implemented in the VASP[51]. An energy cutoff of 520 eV was used and the gamma centered single $k$-point was sampled for integration because of the large cell size. A Grimme's dispersion correction (D3) with a zero damping was also applied[52]. The convergence criteria were 10$^{-6}$ eV and 0.02 eV Å$^{-1}$ for the electronic and ionic cycles, respectively. The monolayer of $Cu_3(HHTP)_2$ was assumed because the long-range order of $Cu_3(HHTP)_2$ has not yet been identified. In order to avoid a fictitious interaction between layers, the vacuum layer along the $z$-direction was set to be ~20 Å so that the lattice size was 21 × 21 × 20 Å. In order to represent the reduction of $Cu_3(HHTP)_2$, we supplied extra electrons to the pristine state and the charge-density difference between the reduced and pristine states was illustrated using the VESTA software[53]. In order to estimate the ion-substitution energy, we employed the hydrated $Zn^{2+}$ and $Cu^{2+}$ states as the reference. To this end, an implicit solvent model[54] was applied and a higher energy cutoff (650 eV) was used.

## Data availability

The authors declare that all the relevant data are available within the paper and its Supplementary Information file or from the corresponding author upon reasonable request.

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

## Acknowledgements

This research was conducted as part of the Joint Center of Excellence in Integrated Nanosystems at King Abdulaziz City for Science and Technology (KACST) and Northwestern University. The authors thank both KACST and NU for their financial support of this research. The Integrated Molecular Structure Education and Research Center (IMSERC) at NU is recognized for the use of its instrumentation. This work made use of the EPIC facility in Northwestern University's NUANCE Center, which receives support from (1) the Soft and Hybrid Nanotechnology Experimental (SHyNE) Resource (NSF ECCS-1542205), (2) the MRSEC program (NSF DMR-1720139) at the Materials Research Center, (3) the International Institute for Nanotechnology (IIN), and (4) the Keck Foundation; and the State of Illinois, through the IIN. C.A.M. and S.S.P. acknowledge support by the Air Force Office of Scientific Research under Award FA9550-17-1-0348 and the National Science Foundation under Grant CHE-1709888. We thank Dr. L. Sun and Dr. M.E. Schott for helpful discussions. The structural study by TEM is partially based on research sponsored by the Air Force Research laboratory under agreement number is FA8650-15-2-5518. The U.S. Government is authorized to repro-duce and distribute reprints for Governmental purposes notwithstanding any copyright notation thereon. The views and conclusions contained herein are those of the authors and should not be interpreted as necessarily representing the official policies or endor-sements, either expressed or implied, of Air Force Research Laboratory or the U.S. Government.

## Author contributions

K.W.N., S.S.P., C.A.M., and J.F.S. designed the research. S.S.P. carried out the synthesis of the active materials. K.W.N. worked on the electrochemical measurements and analysis. R.dR. performed the TEM measurements. H.K. conducted DFT calculations. K.W.N., S.S.P., R.dR., V.P.D., H.K., C.A.M., and J.F.S. co-wrote the paper. J.F.S. supervised the research. All authors discussed the results and commented on the paper.

## Competing interests

The authors declare no competing interests.
