## [Peer Review File · Nature Communications]

Reviewers' comments:

Reviewer #1 (Remarks to the Author):

This paper entitled "Conductive 2D metal-organic framework for high-performance cathodes in aqueous rechargeable zinc batteries" reported a new conductive metal-organic framework (Cu₃(HHTP)₂) as cathode that could deliver a reversible capacity of 228 mAh g⁻¹ at 50 mA g⁻¹. The manuscript is well organized and the experiments are well done. The result and discussion are reasonable and supportive to some extent. However, the mechanism understanding of the Zn²⁺ ion insertion is insufficient. Many improvements are needed in order to meet the high standard of Nature Communications. The manuscript could be acceptable if the following comments can be well addressed and the supplement data can be well provided.

1. The authors claim that the bulk electrical conductivity of 2D metal-organic framework is 0.2 S/cm in terms of literature, how about Cu₃(HHTP)₂-MOF and its electrode in this article? The authors should give some data to indicate their arguments. Few works related to electrical conductivity of 2D materials (Energy Environ. Sci. 2015, 8, 1992-1997; Energy Environ. Sci. 2019, 12, 1780-1804) may be useful for explanation or discussion.

2. The authors should give more explanations about the excellent rate performance of the Cu₃(HHTP)₂-MOF cathode based on the Zn²⁺ ion diffusion coefficient and the conductivity.

3. According to Bragg's equation, the negative shift of diffraction angle in the XRD pattern means the increase of corresponding lattice distance. In page 7, the authors claim that the (100) peak has a slight positive shift. But from figure 5a and 5b, the diffraction angle and lattice distance decreases at same tendency. Please check the structure analysis.

4. In this work, the mass loading of the active material is about 60%; Considering their practical application, the author also should evaluate its cycling performance under high mass loading.

5. The mechanism understanding of the Zn²⁺ ion insertion is insufficient. The authors claim that Zn²⁺ ions are inserted into the pores (about 2 nm) in MOFs in the zinc batteries. Does H₂O insert in Cu₃(HHTP)₂-MOF during insertion of Zn²⁺ ions? Does Cu₃(HHTP)₂-MOF pore size is a major point in increasing battery performance?

Reviewer #2 (Remarks to the Author):

The authors describe a Cu₃(HHTP)₂ 2D conductive MOF that may be utilized as a ZB cathode. The solid-state structure of Cu₃(HHTP)₂, with large pores and high electrical conductivity, provides a dramatically increased rate performance and cyclability compared to those of classical organic-based materials. These results provide key insights into high-performance, 2D conductive MOF designs for battery electrodes. Nevertheless, I do have some suggestions and comments on this manuscript that I think would help to increase the impact of this manuscript. A minor revision is required before it can be accepted for publication.

1. In the abstract, the author mentions that due to the unique structure of the sample, it has better performance as a zinc battery cathode. The author should provide more clearer SEM images and TEM images.

2. What role does HHTP play in Cu₃(HHTP)₂?

3. There is a problem with the expression of redox process in Fig. 1c.

4. The authors have already performed electrochemical tests of Cu₃(HHTP)₂, however, in order to better analyze the electrochemical performance, the authors should provide the simple reaction mechanism.

5. The reference format is not uniform, please check them carefully.

6. Some important articles should be cited. e.g: Advanced Functional Materials, 2018, 28, 1707500; Nano-Micro Lett, 2019, doi: 10.1007/s40820-019-0272-2; Small, 2018, 14, 1803576.

Reviewer #3 (Remarks to the Author):

In this study, the authors provided a Cu₃(HHTP)₂ MOF as cathode material for aqueous Zn ion

battery, large one-dimensional channels and high electrical conductivity is considered to be the main reason for the high battery performance of Zn- Cu₃(HHTP)₂ battery. This new kind of cathode material is interesting, and the new kind of insertion/extraction mechanism is fascinating. However, the explanation for electrochemical reaction is vague and unclear, which makes it insufficient to be published in the top journal of Nat. Commun. The detailed comments are as follows:

- 1) the overall battery reaction formula of Zn- Cu₃(HHTP)₂ battery should be presented.
- 2) The authors say "high bulk electrical conductivity (0.2 S/m)", to our knowledge, this value of electric conductivity is just moderate.
- 3) The authors say "On account of these unique properties, Cu₃(HHTP)₂ shows redox switching at 1.06 V and 0.88 V vs Zn/Zn²⁺ with the highest reversible capacity of 228 mAh g⁻¹ at 50 mA g⁻¹ among those of other MOF-based cathodes for ZBs", in page 3, in which the comparison of the previous researches should be provided.
- 4) The main capacity contribution, capacitive adsorption or diffusion, to the high rate performance of Zn- Cu₃(HHTP)₂ battery is not clear, the authors should provide more evidences. Furthermore, if the capacitive adsorption contributes the main capacity delivery, the authors should give a self-discharge test by resting for 24 h at fully charged state, referring to Nano Energy 56 (2019) 92–99.
- 5) The authors say that XPS results in Figure 4 shows a "redox process between Cu²⁺ and Cu⁺", a more detailed analysis of the XPS data should be conducted. And, in Figure 1c, this redox process of Cu is not seen, which is an obvious logical contradiction.
- 6) Scale bars was not seen in Figure 2d, 2f and 2g, and the TEM image in Figure 2 and Figure 5 are not clear.
- 7) The explanation for electrochemical reaction mechanism during cycling is not clear, a single XPS data is not enough to explain the change of Cu site and semiquinoid component during discharge/charge process. More evidences should be provided, such as DFT calculations, XAS data, etc. Some questions need to be answered, such as, is there some charge transfer process is provided? How the semoquinod changes to hydroquinoid during discharge process? is there H⁺ insertion coexists during the discharge process.
- 8) We doubt that does Zn²⁺ partially replace the Cu²⁺ in the Cu₃(HHTP)₂ framework after long-term cycle, which will lead to a capacity fading problem. The auothers should provide more evidence to prove the structural stability of the Cu₃(HHTP)₂ framework.

Referee: 1

This paper entitled “Conductive 2D metal-organic framework for high-performance cathodes in aqueous rechargeable zinc batteries” reported a new conductive metal-organic framework ($\text{Cu}_3(\text{HHTP})_2$) as cathode that could deliver a reversible capacity of 228 mAh g⁻¹ at 50 mA g⁻¹. The manuscript is well organized and the experiments are well done. The result and discussion are reasonable and supportive to some extent. However, the mechanism understanding of the Zn^{2+} ion insertion is insufficient. Many improvements are needed in order to meet the high standard of Nature Communications. The manuscript could be acceptable if the following comments can be well addressed and the supplement data can be well provided.

Response: We are grateful for the reviewer’s thoughtful comments. In particular, as addressed below, we have proved that Zn^{2+} ions are inserted together with water and, as a consequence, the interfacial resistance is effectively decreased.

1. The authors claim that the bulk electrical conductivity of 2D metal-organic framework is 0.2 S/cm in terms of literature, how about $\text{Cu}_3(\text{HHTP})_2$ -MOF and its electrode in this article? The authors should give some data to indicate their arguments. Few works related to electrical conductivity of 2D materials (Energy Environ. Sci. 2015, 8, 1992-1997; Energy Environ. Sci. 2019, 12, 1780-1804) may be useful for explanation or discussion.

Response: In response to the reviewer’s comments, we measured the bulk electrical conductivities for $\text{Cu}_3(\text{HHTP})_2$ powder and its electrode composite. The bulk electrical conductivities for $\text{Cu}_3(\text{HHTP})_2$ powder and its electrode (two-point probe, pellet) are 0.01 and 0.04 S cm⁻¹, respectively. We have added this new data to the revised version of the manuscript as well as details of the conductivity measurement procedure in the supplementary information.

The value we mentioned, namely 0.2 S m⁻¹, which is for bulk conductivity but it looks as if the unit of measurement is causing confusion. To avoid the confusion, we changed the units and the electrical conductivity measured from a single crystal.

We mentioned that the electrical conductivity is high because, although this value is low compared to ordinary conductive materials, it is high in the context of micro- and

mesoporous materials. In particular, MOFs are typically insulating as a consequence of high electron localization on the organic ligands and the weak hybridization between organic and inorganic moieties. We intended to emphasize that the material itself has conductivity, not the fact it has high conductivity. We agree that the conductivity value is not particularly high, and we have revised manuscript as follows, where highlighting yellow indicates the changed text.

[Page 1]

“... a two-dimensional (2D) conductive metal-organic framework (MOF) with large one-dimensional channels and **high electrical conductivity**,”

[Page 3]

“In addition, their porous structures and **high electrical conductivities** are favorable to ion and electron transport in the framework, improving high rate capability and cyclability.”

“... as the cathode materials for rechargeable aqueous ZBs. **High bulk Electrical conductivity (0.2 S cm⁻¹, four-point probe, single crystal)²⁸** and large pores (~2 nm) facilitate (Fig. 1b) electron and Zn²⁺ ion transport to active sites.”

[Page 4]

The electrical conductivity of Cu₃(HHTP)₂ powder and Cu₃(HHTP)₂ electrode composite (60 wt% Cu₃(HHTP)₂, 20 wt% acetylene black, and 20 wt% PVDF) were measured on a pressed pellet using the two-point probe method. The conductivities obtained were 0.01 and 0.04 S cm⁻¹ for Cu₃(HHTP)₂ powder and electrode composite, respectively. The electrical conductivity of bulk Cu₃(HHTP)₂ electrode matches well the previously reported values²⁹.

[Supplementary Information, Page 2]

Electrical conductivity of Cu₃(HHTP)₂ was measured by the two-point probe method at 25 °C. A pellet was placed on a home-built in situ pellet press² and connected to an electrometer (Keithley 4200-SCS). The current-voltage (I-V) measurements were performed at 25 °C by sweeping the voltage.

2. The authors should give more explanations about the excellent rate performance of the Cu₃(HHTP)₂-MOF cathode based on the Zn²⁺ ion diffusion coefficient and the conductivity.

Response: This suggestion is a very good one. We measured the Zn²⁺ ion-diffusion coefficient using the galvanostatic intermittent titration technique (GITT) and the conductivity of Zn²⁺ ion at the interface between electrode and electrolytes using the EIS measurement method. The high rate performance of Cu₃(HHTP)₂ is mainly a consequence of the high diffusion rate of Zn²⁺ ions in the host structure and low interfacial resistance, attributed to the low desolvation energy (co-insertion of water with Zn²⁺ ions). We have now added the new section “Origin of high rate performance of Cu₃(HHTP)₂” and discussed the GITT and EIS experiments in the revised manuscript and supplementary information as follows:

[Page 6–8]

Origin of high rate performance of Cu₃(HHTP)₂

[Page 12]

“...with high diffusion rate of Zn²⁺ ions, and low interfacial resistance caused by the insertion of hydrated Zn²⁺ ions, as a result of the large open channel structures provides a dramatically increased rate performance and cyclability compared to those of classical organic-based materials.”

[Page 14]

EIS measurements were performed on symmetric cells over the frequency range of 0.01 Hz–1 MHz with an input voltage amplitude of 10 mV (Reference 600 potentiostat, Gamry Instruments, USA).

[Supplementary Information, Page 5]

The chemical diffusion coefficient ($D_{Zn^{2+}}$) presented here is the average value of all D calculated in each interval according to Equation (S1, below), which was originally derived by Weppner and Huggins⁹.

$$D = \frac{4}{\pi\tau} \left(\frac{m_B V_M}{M_B S} \right)^2 \left(\frac{\Delta E_s}{\Delta E_t} \right)^2 (\tau \ll L^2/D) \quad (S1)$$

τ : time for an applied galvanostatic current I_0

m_B : mass of active material

V_m : molar volume of active material

M_B : molecular weight of active material

S : interface area between the active material and electrolyte

ΔE_s : steady-state (equilibrium) voltage

ΔE_t : total change of the cell voltage E during the current pulse

L : radius of the active particle

[Added Supplementary Fig. 4] A GITT curve of $\text{Cu}_3(\text{HHTP})_2$ for the first charge process between 0.5 V and 1.3 V (current density: 50 mA g^{-1} , time interval τ : 30 min).

[Added Supplementary Fig. 5] EIS spectra for $\text{Cu}_3(\text{HHTP})_2$ electrodes in (a) an aqueous and (b) an organic electrolyte. 3 M $\text{Zn}(\text{CF}_3\text{SO}_3)_2$ in DI H_2O and 0.25 M $\text{Zn}(\text{CF}_3\text{SO}_3)_2$ in MeCN were used as the electrolytes for aqueous and organic electrolytes, respectively.

3. According to Bragg's equation, the negative shift of diffraction angle in the XRD pattern means the increase of corresponding lattice distance. In page 7, the authors claim that the (100) peak has a slight positive shift. But from figure 5a and 5b, the diffraction angle and lattice distance decreases at same tendency. Please check the structure analysis.

Response: We thank the reviewer for pointing out this mistake. The color of the legend in Fig. 5a was incorrect. We have now changed the color of the legend.

[Revised Fig. 5a]

4. In this work, the mass loading of the active material is about 60%; Considering their practical application, the author also should evaluate its cycling performance under high mass loading.

Response: This suggestion is another welcome one from the reviewer. We have evaluated the cycling performance of $\text{Cu}_3(\text{HHTP})_2$ with 90% active materials. Although the initial

capacity of $\text{Cu}_3(\text{HHTP})_2$ with 90% active materials decreased slightly, the cycle retention was well maintained. We have now added the Figures to the supplementary information along with discussions about the results in the revised manuscript as follows:

[Page 6]

In addition, by increasing the mass loading of active materials from 60 to 90%, although the initial capacity decreased slightly to 125 mAh g^{-1} at 500 mA g^{-1} (Supplementary Fig. 3a), the retention of capacity after 100 cycles was 76% of the initial capacity (Supplementary Fig. 3b). This capacity retention of 90% active materials loading electrode is almost identical to that of a 60% active materials loading electrode.

[Page 14]

“... by making a slurry containing $\text{Cu}_3(\text{HHTP})_2$: acetylene black : poly(vinylidene difluoride) (PVDF) in the ratio of 60 : 20 : 20 or 90 : 5 : 5 in 1-methyl-2-pyrrolidinone (NMP), respectively.”

[Added Supplementary Fig. 3] (a) Discharge–charge voltage profiles and (b) cycling performance of $\text{Cu}_3(\text{HHTP})_2$, depending on the mass loading of active materials at 500 mA g^{-1} . The electrode with 90% active materials was composed of $\text{Cu}_3(\text{HHTP})_2$: acetylene black : PVDF = 90 : 5 : 5.

5. The mechanism understanding of the Zn^{2+} ion insertion is insufficient. The authors claim that Zn^{2+} ions are inserted into the pores (about 2 nm) in MOFs in the zinc batteries. Does H_2O insert in $\text{Cu}_3(\text{HHTP})_2$ -MOF during insertion of Zn^{2+} ions? Dose $\text{Cu}_3(\text{HHTP})_2$ -MOF pore size is a major point in increasing battery performance?

Response: By using TGA analysis, we have confirmed that H₂O is inserted along with Zn²⁺ ions during the discharge process. The insertion of H₂O with Zn²⁺ ions effectively decreases the interfacial resistance because the desolvation process of H₂O can, in part, be omitted (R1–R4). In order to understand the H₂O effects on battery performance, more clearly, we have also evaluated the electrochemical performance of Cu₃(HHTP)₂ in the organic electrolytes and confirmed that the initial capacity and the cycle retention of Cu₃(HHTP)₂ in the organic electrolytes are much poorer than those in the aqueous electrolytes. Based on these results, we can assume that the large pore size of Cu₃(HHTP)₂ supports the insertion of hydrated Zn²⁺ ions. We have added a new section, “Origin of high rate performance of Cu₃(HHTP)₂”, based on these experimental results, as well as revised, added figures and relevant text as follows:

(R1) Okoshi, M., Yamada, Y., Yamada, A. & Nakai, H. Theoretical analysis on desolvation of lithium, sodium, and magnesium cations to organic electrolyte solvents. *J. Electrochem. Soc.* **160**, A2160–A2165 (2013).

(R2) Mizuno, Y. *et al.* Suppressed activation energy for interfacial charge transfer of a prussian blue analog thin film electrode with hydrated ions (Li⁺, Na⁺, and Mg²⁺). *J. Phys. Chem. C* **117**, 10877–10882 (2013).

(R3) Nam, K. W. *et al.* The high performance of crystal water containing manganese birnessite cathodes for magnesium batteries. *Nano Lett.* **15**, 4071–4079 (2015).

(R4) Nam, K. W. *et al.* Crystal water for high performance layered manganese oxide cathodes in aqueous rechargeable zinc batteries. *Energy Environ. Sci.* **12**, 1999–2009 (2019).

[Page 6–8]

Origin of high rate performance of Cu₃(HHTP)₂

[Page 12]

In addition, the kinetic analyses of the electrochemical behavior of Cu₃(HHTP)₂ obtained by CV suggest that the charge-storage mechanism of Cu₃(HHTP)₂ is intercalation pseudocapacitance, indicating that the mechanism is not determined by diffusion.

[Page 17]

35. Okoshi, M., Yamada, Y., Yamada, A. & Nakai, H. Theoretical analysis on de-solvation of lithium, sodium, and magnesium cations to organic electrolyte solvents. *J. Electrochem. Soc.* **160**, A2160–A2165 (2013).

36. Mizuno, Y. *et al.* Suppressed activation energy for interfacial charge transfer of a prussian blue analog thin film electrode with hydrated ions (Li^+ , Na^+ , and Mg^{2+}). *J. Phys. Chem. C* **117**, 10877–10882 (2013).

37. Nam, K. W. *et al.* The high performance of crystal water containing manganese birnessite cathodes for magnesium batteries. *Nano Lett.* **15**, 4071–4079 (2015).

38. Nam, K. W., Kim, H., Choi, J. H. & Choi, J. W. Crystal water for high performance layered manganese oxide cathodes in aqueous rechargeable zinc batteries. *Energy Environ. Sci.* **12**, 1999–2009 (2019).

[Supplementary Information, Page 2]

“... an electron beam energy of 20 kV and emission current of 20 μA . In order to investigate the H_2O content after the discharge process, thermogravimetric analysis (TGA, Netzsch Jupiter) was performed by raising the temperature from room temperature to 300 $^\circ\text{C}$ at a ramping rate of 5 $^\circ\text{C min}^{-1}$ under an Ar flow.”

[Revised Fig. 1]

[redacted]

[Added Supplementary Fig. 6] **(a)** Discharge–charge voltage profiles and **(b)** cycling performance of $\text{Cu}_3(\text{HHTP})_2$ with 0.25 M $\text{Zn}(\text{CF}_3\text{SO}_3)_2$ in MeCN at 50 mA g^{-1}

[Added Supplementary Fig. 11] TGA profiles of $\text{Cu}_3(\text{HHTP})_2$ electrodes at the pristine and the discharged states

Referee: 2

The authors describe a $\text{Cu}_3(\text{HHTP})_2$ 2D conductive MOF that may be utilized as a ZB cathode. The solid-state structure of $\text{Cu}_3(\text{HHTP})_2$, with large pores and high electrical conductivity, provides a dramatically increased rate performance and cyclability compared to those of classical organic-based materials. These results provide key insights into high-performance, 2D conductive MOF designs for battery electrodes. Nevertheless, I do have some suggestions and comments on this manuscript that I think would help to increase the impact of this manuscript. A minor revision is required before it can be accepted for publication.

Response: We thank the reviewer for this positive evaluation.

1. In the abstract, the author mentions that due to the unique structure of the sample, it has better performance as a zinc battery cathode. The author should provide more clearer SEM images and TEM images.

Response: As the reviewer points out, we have added SEM images and changed TEM images as indicated below:

[Added Supplementary Fig. 8] SEM images of $\text{Cu}_3(\text{HHTP})_2$ powder at (a, b) low and (c) high resolution. (d) EDX spectrum for the selected area in c. Scale bars in a-c are 500, 250, and 100 nm, respectively.

[Revised Figs. 2c and 2d]

[Revised Supplementary Fig. 7]

[Revised Fig. 5c]

2. What role does HHTP play in $\text{Cu}_3(\text{HHTP})_2$?

Response: Hexahydroxytriphenylene (HHTP), which interconverts reversibly between the semiquinonate and quinone forms, provides redox activity to the material as well as imparting a square-planar coordination geometry with Cu(II). The structure ends up being a 2D-honeycomb one. In order to make this point clear, we have revised the manuscript as follows:

[Page 3]

“In particular, we anticipate that the redox activity of the quinoid units of HHTP^{28–30}, with Zn^{2+} insertion, will promote (Fig. 1c) the performance of the cathode.”

3. There is a problem with the expression of redox process in Fig. 1c.

Response: We thank the reviewer for pointing out this situation. We have revised Fig. 1c, including the redox reaction of copper.

[Revised Fig. 1c]

4. The authors have already performed electrochemical tests of $\text{Cu}_3(\text{HHTP})_2$, however, in order to better analyze the electrochemical performance, the authors should provide the simple reaction mechanism.

Response: The reaction mechanism of $\text{Cu}_3(\text{HHTP})_2$ is an intercalation pseudocapacitance charge-storage mechanism. We have added a new section, “Charge-storage mechanism of $\text{Cu}_3(\text{HHTP})_2$ ”, and included more details in the discussion.

[Page 11–12]

Charge-storage mechanism of $\text{Cu}_3(\text{HHTP})_2$

5. The reference format is not uniform, please check them carefully.

Response: We appreciate the reviewer for pointing out this fact. We have checked the references thoroughly are now all in a uniform format in the revised manuscript.

6. Some important articles should be cited. e.g: *Advanced Functional Materials*, 2018, 28, 1707500; *Nano-Micro Lett*, 2019, doi: 10.1007/s40820-019-0272-2; *Small*, 2018, 14, 1803576.

Response: We thank reviewer for these good suggestions. We have now cited the suggested references in the revised manuscript.

[Page 15–17]

6. Zheng, M. *et al.* Tungsten-based materials for lithium-ion batteries. *Adv. Func. Mater.* **28**, 1707500 (2018).

7. Zhou, H., Li, X., Li, Y., Zheng, M. & Pang, H. Applications of M_xSe_y (M = Fe, Co, Ni) and their composites in electrochemical energy storage and conversion. *Nano-Micro Lett.* **11**, 40 (2019).

30. Zhu, R. *et al.* π -Conjugated molecule boosts metal-organic frameworks as efficient oxygen evolution reaction catalysts. *Small* **14**, 1803576 (2018).

Referee: 3

In this study, the authors provided a $Cu_3(HHTP)_2$ MOF as cathode material for aqueous Zn ion battery, large one-dimensional channels and high electrical conductivity is considered to be the main reason for the high battery performance of Zn- $Cu_3(HHTP)_2$ battery. This new kind of cathode material is interesting, and the new kind of insertion/extraction mechanism is fascinating. However, the explanation for electrochemical reaction is vague and unclear, which makes it insufficient to be published in the top journal of Nat. Commun. The detailed comments are as follows:

Response: We thank reviewer for the detailed comments. We have performed many experiments to explain the electrochemical reaction. With the newly inserted corrections, we hope that the manuscript is now suitable for publication in *Nature Communications*.

1) The overall battery reaction formula of Zn-Cu₃(HHTP)₂ battery should be presented.

Response: The initial reversible capacity of Cu₃(HHTP)₂ is 228 mAh g⁻¹ at a rate of 50 mA g⁻¹ (see Fig. 3a), indicating that the repeating unit in Fig. 1c obtains 2.3 electrons and 1.15 Zn²⁺ while discharging. Based on the XPS Cu 2*p* (Fig. 4c) and the capacity loss after a self-discharge test (inset of the Fig. 6d), we assume that the additional capacity of 39 mAh g⁻¹ comes from the redox region of Cu²⁺/Cu⁺. This capacity of 39 mA g⁻¹ corresponds to accepting about 0.3 electrons, and thus the 0.3 Cu²⁺ in the repeating unit in its pristine state (left side, Fig. 1c) is reduced to 0.3 Cu⁺ at the discharged state (right side, Fig. 1c). Because Cu₃(HHTP)₂ consists of three repeating units of the structural formula in Fig. 1c, the overall reaction formula of the Zn-Cu₃(HHTP)₂ battery is Cu₃(HHTP)₂ + 3.45Zn²⁺ + 6.9e⁻ → Zn_{3.45}[Cu₃(HHTP)₂]. We have now included this information in the revised manuscript and supplementary information:

[Page 9]

In light of these XPS results, the additional discharge capacity of Cu₃(HHTP)₂, equivalent to 0.3 electrons, can be derived from the redox events of Cu²⁺.

[Supplementary Information, Page 11]

Section F. Overall Reaction Formula of Zn-Cu₃(HHTP)₂ Battery

The initial reversible capacity of Cu₃(HHTP)₂ is 228 mAh g⁻¹ at a rate of 50 mA g⁻¹ (see Fig. 3a), indicating that the repeating unit in Fig. 1c obtains 2.3 electrons and 1.15 Zn²⁺ while discharging. Based on the XPS Cu 2*p* (Fig. 3c) and the capacity loss after a self-discharge test (inset of Fig. 6d), we assume that the additional capacity of 39 mAh g⁻¹ comes from the redox region of Cu²⁺/Cu⁺. This capacity of 39 mA g⁻¹ corresponds to accepting about 0.3 electrons, and thus the 0.3 Cu²⁺ in the repeating unit in its pristine state (left side, Fig. 1c) reduced at the discharged state (right side, Fig. 1c). Because Cu₃(HHTP)₂ consists of three repeating units of the structural formula in Fig. 1c, the overall reaction formula of Zn-Cu₃(HHTP)₂ battery is as follows:

2) The authors say “high bulk electrical conductivity (0.2 S/m)”, to our knowledge, this value of electric conductivity is just moderate.

We agreed with the reviewer that the conductivity value is not unusually high. We mentioned that the electrical conductivity is high because, although this value is low compared to ordinary conductive materials, it is high compared to micro- and mesoporous materials. In the case of MOFs, for example, they are typically insulating as a consequence of high electron localization on the organic ligand and weak hybridization between the organic and inorganic moieties. In other words, we intended to emphasize that the material itself has conductivity, not the fact that it has high conductivity. To correct the misleading statements and provide more information about the conductivity of the material itself, rather than the bulk conductivity, we have revised the manuscript as follows:

[Page 1]

“... a two-dimensional (2D) conductive metal-organic framework (MOF) with large one-dimensional channels and high electrical conductivity,”

[Page 3]

“In addition, their porous structures and high electrical conductivities are favorable to ion and electron transport in the framework, improving high rate capability and cyclability.”

“... as the cathode materials for rechargeable aqueous ZBs. ~~High bulk~~ Electrical conductivity (0.2 S cm⁻¹, four-point probe, single crystal)²⁸ and large pores (~2 nm) facilitate (Fig. 1b) electron and Zn²⁺ ion transport to active sites.”

[Page 4]

The electrical conductivity of Cu₃(HHTP)₂ powder and Cu₃(HHTP)₂ electrode composite (60 wt% Cu₃(HHTP)₂, 20 wt% acetylene black, and 20 wt% PVDF) were measured on a pressed pellet using the two-point probe method. The conductivities obtained were 0.01 and 0.04 S cm⁻¹ for Cu₃(HHTP)₂ powder and electrode composite, respectively. The electrical conductivity of bulk Cu₃(HHTP)₂ electrode matches well the previously reported values²⁹.

3) The authors say “On account of these unique properties, $\text{Cu}_3(\text{HHTP})_2$ shows redox switching at 1.06 V and 0.88 V vs Zn/Zn^{2+} with the highest reversible capacity of 228 mAh g^{-1} at 50 mA g^{-1} among those of other MOF-based cathodes for ZBs”, in page 3, in which the comparison of the previous researches should be provided.

Response: We thank reviewer for pointing out this omission. We have now added sentences about the comparison with the previous research in the revised manuscript as follows:

[Page 3]

“...On account of these unique properties, $\text{Cu}_3(\text{HHTP})_2$ shows redox switching at 1.06 V and 0.88 V vs Zn/Zn^{2+} with the highest reversible capacity of 228 mAh g^{-1} at 50 mA g^{-1} . These reversible capacities in rechargeable aqueous ZBs are the first example in MOFs and the highest reported values for cathodes with open-framework structures, including Prussian Blue analogues^{31–33} that have exhibited substantially smaller values of < 70 mAh g^{-1} at similar current densities.”

4) The main capacity contribution, capacitive adsorption or diffusion, to the high rate performance of $\text{Zn}-\text{Cu}_3(\text{HHTP})_2$ battery is not clear, the authors should provide more evidences. Furthermore, if the capacitive adsorption contributes the main capacity delivery, the authors should give a self-discharge test by resting for 24 h at fully charged state, referring to Nano Energy 56 (2019) 92–99.

Response: We thank the reviewer for suggesting these experiments. In order to understand the charge-storage mechanism, CV measurements were carried out at various different scan rates. Current values, depending on the scan rate, enabled us to determine *b*-values which are above 0.85 within all operating voltage ranges, indicating an operating mechanism not dominated by diffusion. We also tried a self-discharge test and observed a self-discharge rate of 0.003 Vh^{-1} and 83% of initial capacity being maintained after 5 days. We have included this new data and also more details in the discussion in a new section, “Charge-storage mechanism of $\text{Cu}_3(\text{HHTP})_2$ ”; the following references and a Figure are included in this section, as follows:

[Page 11–12]

Charge-storage mechanism of $\text{Cu}_3(\text{HHTP})_2$

[Page 17–18]

39. Wang, J., Polleux, J., Lim, J. & Dunn, B. Pseudocapacitive contributions to electrochemical energy storage in TiO_2 (Anatase) nanoparticles. *J. Phys. Chem. C* **111**, 14925–14931 (2007).

40. Augustyn, V. *et al.* High-rate electrochemical energy storage through Li^+ intercalation pseudocapacitance. *Nat. Mater.* **12**, 518–522 (2013).

41. Brezesinski, K. *et al.* Pseudocapacitive contributions to charge storage in highly ordered mesoporous group V transition metal oxides with iso-oriented layered nanocrystalline domains. *J. Am. Chem. Soc.* **132**, 6982–6990 (2010).

42. Brezesinski, T., Wang, J., Tolbert, S. H. & Dunn, B. Ordered mesoporous $\alpha\text{-MoO}_3$ with iso-oriented nanocrystalline walls for thin-film pseudocapacitors. *Nat. Mater.* **9**, 146–151 (2010).

43. Zukalová, M., Kalbáč, M., Kavan, L., Exnar, I. & Graetzel, M. Pseudocapacitive lithium storage in $\text{TiO}_2(\text{B})$. *Chem. Mater.* **17**, 1248–1255 (2005).

44. Come, J. *et al.* Electrochemical kinetics of nanostructured Nb_2O_5 electrodes. *J. Electrochem. Soc.* **161**, A718–A725 (2014).

45. Wang, Z. *et al.* A MOF-based single-ion Zn^{2+} solid electrolyte leading to dendrite-free rechargeable Zn batteries. *Nano Energy* **56**, 92–99 (2019).

[Added Fig. 6] **Charge-storage mechanism of $\text{Cu}_3(\text{HHTP})_2$.** **a**, Cyclic voltammograms of $\text{Cu}_3(\text{HHTP})_2$ recorded at different scan rates. **b**, b -values for the $\text{Cu}_3(\text{HHTP})_2$ electrodes plotted as a function of the potential for cathodic scans. **c**, Capacitive and diffusion currents contributed to the charge-storage of $\text{Cu}_3(\text{HHTP})_2$ at the rate of 0.5 mV s^{-1} . **d**, A self-discharge profile of $\text{Cu}_3(\text{HHTP})_2$. The inset shows voltage profiles before and after storage for the self-discharge test.

[Added Supplementary Fig. 14] **(a)** Peak currents as a function of scan rate in the range of 0.6–1.2 V during reduction process. **(b)** Use of equation S3 to analyze the capacitance currents of $\text{Cu}_3(\text{HHTP})_2$ electrode. The solid lines are visual guides to the experimental data points.

5) The authors say that XPS results in Figure 4 shows a “redox process between Cu^{2+} and Cu^{+} ”, a more detailed analysis of the XPS data should be conducted. And, in Figure 1c, this redox process of Cu is not seen, which is an obvious logical contradiction.

Response: We appreciate the reviewer's comment and have revised Fig. 1c to include the redox reaction of copper. Furthermore, we think that the redox reaction going from Cu^{2+} to Cu^+ is clearly shown in Fig. 4c, because the peak for the Cu^{2+} satellite disappears and the Cu $2p_{1/2}$ and Cu $2p_{3/2}$ peaks are shifted to lower binding energy after passing the first plateau around 0.9 V (vs. Zn/Zn^{2+}) in Fig. 3a. Based on the standard electrode potentials in aqueous solution at 25 °C, the potential at which Cu^{2+} to Cu^+ occurs is 0.16 V (vs. SHE), which can be transformed to 0.92 V (vs. Zn/Zn^{2+}).

In addition, the redox reaction from Cu^{2+} to Cu^+ in Cu(2,7-AQDC) MOF also occurred around 3.1 V (vs. Li/Li^+): if can be transformed to 0.86 V (vs. Zn/Zn^{2+}) (R5). Consequently, based on our findings and previous results, the first plateau of $\text{Cu}_3(\text{HHTP})_2$ relates to the redox reaction from Cu^{2+} to Cu^+ . Furthermore, we performed DFT calculations to gain a better understanding of the redox behavior of $\text{Cu}_3(\text{HHTP})_2$. More discussion about the DFT calculations is included in the response to question 7. Also, we revised Fig. 1c to show Cu is involved in the redox process.

(R5) Zhang, Z. *et al.* Monitoring the solid-state electrochemistry of Cu(2,7-AQDC) (AQDC = anthraquinone dicarboxylate) in a lithium battery: Coexistence of metal and ligand redox activities in a metal-organic framework. *J. Am. Chem. Soc.* **136**, 16112–16115 (2014).

[Revised Fig. 1c]

6) Scale bars was not seen in Figure 2d, 2f and 2g, and the TEM image in Figure 2 and Figure 5 are not clear.

Response: We appreciate the reviewer for pointing out this lack of clarity. We have added the scale bars in Fig. 2d, 2f, and 2g. In addition, we have revised the TEM images in Fig. 2c, Fig. 2d and Fig. 5c.

7) The explanation for electrochemical reaction mechanism during cycling is not clear, a single XPS data is not enough to explain the change of Cu site and semiquinoid component during discharge/charge process. More evidences should be provided, such as DFT calculations, XAS data, etc. Some questions need to be answered, such as, is there some charge transfer process is provided? How the semoquinod changes to hydroquinoid during discharge process? is there H^+ insertion coexists during the discharge process.

Response: We thank to reviewer for pointing out the confusion. The change on going from the semiquinoid to the hydroquinoid does not mean the insertion of H^+ ions into the $Cu_3(HHTP)_2$. We agree the naming was misleading to readers. We changed the names to quinoid and benzoid instead of semiquinoid and hydroquinoid in the revised manuscript. No hydrogen insertion coexists during the process.

We conducted DFT calculations (Fig. 4d) in order to obtain a clearer understanding of atomic sites involved in the redox reaction of $Cu_3(HHTP)_2$. Fig. 4d shows an increment of electron density after the reduction of $Cu_3(HHTP)_2$. The change in oxygen was most pronounced and electrons were also placed at copper and linker atoms. It indicates that copper, as well as oxygen, participates in the redox reaction. In addition, we identified that the electronic band of $Cu_3(HHTP)_2$ (Supplementary Fig. 12) just above the Fermi level (which corresponds to the LUMO) is composed of C, O, and Cu, supporting the notion that these atoms can be reduced upon Zn^{2+} insertion. DFT results have been included in the revised manuscript and a detailed procedure for performing the calculations is in the revised supplementary information as follows:

[Page 9]

In order to identify the redox center of $Cu_3(HHTP)_2$, density functional theory (DFT) calculations were performed. When supplying 6.9 extra electrons to $Cu_3(HHTP)_2$, Cu atom, as well as to the linker, takes of the additional electron (Fig. 4d), indicating Cu atoms participate in the reduction reaction. In density of states (DOS) analysis (Supplementary Fig. 12), the electronic states just above the Fermi level consist of O, C, and Cu. This result supports the observed redox events occurring at these atoms.

[Supplementary Information, Page 3]

DFT calculations were performed using the Perdew-Burke-Ernzhofer (PBE) exchange-correlation functional³ and the projector-augmented wave (PAW) method⁴ as implemented in the VASP⁵. An energy cutoff of 520 eV was used and the gamma centered single k -point was sampled for integration because of the large cell size. A Grimme's dispersion correction (D3) with a zero damping was also applied⁶. The convergence criteria were 10^{-6} eV and 0.02 eV \AA^{-1} for the electronic and ionic cycles, respectively. The monolayer of $\text{Cu}_3(\text{HHTP})_2$ was assumed because the long-range order of $\text{Cu}_3(\text{HHTP})_2$ has not yet been identified. In order to avoid a fictitious interaction between layers, the vacuum layer along the z -direction was set to be $\sim 20 \text{ \AA}$ so that the lattice size was $21 \times 21 \times 20 \text{ \AA}$. In order to represent the reduction of $\text{Cu}_3(\text{HHTP})_2$, we supplied extra electrons to the pristine state; and the charge-density difference between the reduced and pristine states was illustrated using the VESTA software⁷. In order to estimate the ion-substitution energy, we employed the hydrated Zn^{2+} and Cu^{2+} states as the reference. To this end, an implicit solvent model⁸ was applied and a higher energy cutoff (650 eV) was used.

[Added Supplementary Fig. 12] Density of states near the Fermi level of the $\text{Cu}_3(\text{HHTP})_2$ monolayer.

[Revised Fig. 4] **Electronic states analysis during discharge–charge. a-c**, Ex situ XPS spectra of **(a)** Zn 2*p*, **(b)** O 1*s*, and **(c)** Cu 2*p*. **d**, Changes of electron density upon the reduction of Cu₃(HHTP)₂.

8) We doubt that does Zn²⁺ partially replace the Cu²⁺ in the Cu₃(HHTP)₂ framework after long-term cycle, which will lead to a capacity fading problem. The authors should provide more evidence to prove the structural stability of the Cu₃(HHTP)₂ framework.

Response: We think that the structural stability of the Cu₃(HHTP)₂ – analyzed by ex situ XRD and XPS – after 500 cycles of charging/discharging at 4000 mA g⁻¹ supports the fact that the crystal structure and electronic structure of the Cu₃(HHTP)₂ are well maintained after long-term cycling. Also, according to DFT calculations, ion-exchange from Cu²⁺ to Zn²⁺ ions was endothermic by 0.8 eV per ion (Supplementary Fig. 13). This result supports the high stability of Cu₃(HHTP)₂ against the Zn substitution. We have discussed these results in the revised manuscript and added details in the supplementary information as follows:

[Page 10]

In addition, ion-exchange from Cu^{2+} to Zn^{2+} ions is endothermic by 0.8 eV per ion, according to DFT calculations (Supplementary Fig. 13), indicating the high stability of $\text{Cu}_3(\text{HHTP})_2$ against Zn^{2+} substitution.

[Added Supplementary Fig. 13] Changes of relative energy after substituting Cu^{2+} to Zn^{2+} ions in $\text{Cu}_3(\text{HHTP})_2$. As a reference state for energy comparison, respective hydrated divalent cations with an implicit solvent model were used.

REVIEWERS' COMMENTS:

Reviewer #1 (Remarks to the Author):

The authors have addressed the related concerns and the manuscript improved a lot, it is recommended for publication.

Reviewer #2 (Remarks to the Author):

Accept

Reviewer #3 (Remarks to the Author):

I recommend this article to be published in Nat. Commun. after tackling the concern below: In response for question 7, we wonder if the H⁺ insertion coexists during the discharge process, but the authors did not give a direct explanation. In fact, it's very simple to identify if the H⁺ insertion exists. If there has a H⁺ insertion, zinc hydroxide hydrate analogue compound will form on the electrode surface after discharging, according to the reference of Nat. Commun. 2018, 9, 2906. If this part of the results can be supplemented, the related charge/discharge mechanism for this MOF cathode will be complete.

Referee: 1

The authors have addressed the related concerns and the manuscript improved a lot, it is recommended for publication.

Response: We thank Referee 1 for his/her appreciation of this work.

Referee: 2

Accept

Response: We thank Referee 2 for his/her appreciation of this work.

Referee: 3

I recommend this article to be published in Nat. Commun. after tackling the concern below:

In response for question 7, we wonder if the H⁺ insertion coexists during the discharge process, but the authors did not give a direct explanation. In fact, it's very simple to identify if the H⁺ insertion exists. If there has a H⁺ insertion, zinc hydroxide hydrate analogue compound will form on the electrode surface after discharging, according to the reference of Nat. Commun. 2018, 9, 2906. If this part of the results can be supplemented, the related charge/discharge mechanism for this MOF cathode will be complete.

Response: We appreciated the Referee's suggestion. According to the suggested reference, the strong emerging XRD peaks at 8.1°, 16.2°, and 24.4° for the zinc hydroxide hydrated analogue are observed because of the H⁺ ions insertion (R1). In the XRD peak of Cu₃(HHTP)₂ at the discharged state (Fig. 5a), however, the main peaks related to the zinc hydroxide hydrate analogue are not observed. Therefore, the Cu₃(HHTP)₂ does not include the insertion of H⁺ ions during discharge reaction. We have discussed this point in the revised manuscript and changed Fig. 5a to show that there are no changes in peaks – more obviously in the wider range – and added the reference as follows:

(R1) Huang, J. *et al.* Polyaniline-intercalated manganese dioxide nanolayers as a high-performance cathode material for an aqueous zinc-ion battery. *Nat. Commun.* **9**, 2906 (2018).

[Page 10]

With the exception of peak shifts following Zn^{2+} insertion, no changes (appearance or disappearance of peaks) are observed, indicating that the discharge process does not include H^+ insertion accompanied by the formation of the $\text{Zn}(\text{OH})_2$ analogue³⁹.

[Page 19]

39. Huang, J. *et al.* Polyaniline-intercalated manganese dioxide nanolayers as a high-performance cathode material for an aqueous zinc-ion battery. *Nat. Commun.* **9**, 2906 (2018).

[Revised Fig. 5a]